# Predictors of Abdominal Aortic Aneurysm Risks

**DOI:** 10.3390/bioengineering7030079

**Published:** 2020-07-22

**Authors:** Stephen J. Haller, Amir F. Azarbal, Sandra Rugonyi

**Affiliations:** 1Department of Biomedical Engineering, Oregon Health & Science University, Portland, OR 97239, USA; stephen.haller@unmc.edu; 2Division of Vascular Surgery, Oregon Health & Science University, Portland, OR 97239, USA; azarbala@ohsu.edu

**Keywords:** abdominal aneurysm, aortic aneurysm, aorta biomechanics, risk assessment, rupture potential index, aortic wall stress

## Abstract

Computational biomechanics via finite element analysis (FEA) has long promised a means of assessing patient-specific abdominal aortic aneurysm (AAA) rupture risk with greater efficacy than current clinically used size-based criteria. The pursuit stems from the notion that AAA rupture occurs when wall stress exceeds wall strength. Quantification of peak (maximum) wall stress (PWS) has been at the cornerstone of this research, with numerous studies having demonstrated that PWS better differentiates ruptured AAAs from non-ruptured AAAs. In contrast to wall stress models, which have become progressively more sophisticated, there has been relatively little progress in estimating patient-specific wall strength. This is because wall strength cannot be inferred non-invasively, and measurements from excised patient tissues show a large spectrum of wall strength values. In this review, we highlight studies that investigated the relationship between biomechanics and AAA rupture risk. We conclude that combining wall stress and wall strength approximations should provide better estimations of AAA rupture risk. However, before personalized biomechanical AAA risk assessment can become a reality, better methods for estimating patient-specific wall properties or surrogate markers of aortic wall degradation are needed. Artificial intelligence methods can be key in stratifying patients, leading to personalized AAA risk assessment.

## 1. Introduction

Abdominal aortic aneurysm (AAA) is a potentially life-threatening condition, characterized as a pathological expansion of the abdominal aorta, wherein the maximal transverse diameter exceeds 30 mm [1,2,3]. Aneurysm rupture, by definition, occurs when aortic wall stress exceeds aortic wall strength [4,5], which is the stress/tension at which the wall can no longer withstand the forces applied to it (e.g., due to blood pressure). Rupture represents the main concern associated with AAA, as AAA rupture carries an overall mortality rate of approximately 80–85% [6,7]. Indeed, recent studies (using data ranging from 1995 to 2013) report a one-year mortality rate of hospitalized patients with a ruptured AAA to be about 40% [8,9]. Nevertheless, it is estimated that between 50 and 66% of patients do not make it to the hospital alive [10,11], making the overall mortality rate for ruptured AAA between 70–80%, even in modern times. Because most patients with AAAs are largely asymptomatic prior to rupture, effective means for screening and evaluation are paramount for proper AAA detection and treatment. Nowadays, ultrasound is the preferred imaging modality for both AAA diagnosis and monitoring AAA progression, although computer tomography (CT) and magnetic resonance imaging (MRI) are also employed [10].

Surgical criteria for elective AAA repair should carefully weigh the risk of rupture versus the risk of repair. Surgery has an in-hospital mortality rate of approximately 3–5% for open repair and 1–2% for endovascular repair (EVAR) [12,13], thus, EVAR (which is also less invasive) has been the preferred repair procedure. Four recent clinical trials in Europe and the US (EVAR-1, DREAM, OVER, ACE), in which patients undergoing AAA elective repair were randomized for open repair or EVAR, have indeed shown the early survival advantage of EVAR [14]. However, this early advantage was lost after three years post intervention, mainly due to complications such as secondary rupture and re-intervention in the EVAR group [14,15]. After 12 years follow-up, survival rates dropped to about 40% for both groups, with rates of re-intervention that continued to be higher for EVAR [16]. While early (30 day) in-hospital mortality rates are about 3.4% for elective repair (open surgery and EVAR combined) and 33% for repair after rupture, there is no survival benefit or increased mortality of emergency versus elective repair after 30 days [17]. Clinical studies have long ago determined that the risk of repair is exceeded by the risk of rupture when either: (1) AAA maximal transverse diameter is greater than 55 mm for men or 50 mm for women; or (2) annual AAA growth rate is greater than 10 mm/year [18,19,20,21]. These criteria have not changed in over 20 years. Unfortunately, some patients with AAAs that do not meet these criteria may still rupture. The five-year cumulative rate of rupture of aneurysms smaller than 5 cm is 1–7% [10], with annual rupture risk for AAAs between 40 and 55 mm in diameter approximately 1% [22]. Autopsy studies, however, have suggested this rate may be even higher [23,24,25,26]. However, recent studies have shown no advantage for early AAA repair [27,28]. Moreover, large AAAs may not all be equally prone to rupture, with a five-year cumulative rate of rupture of 25–40% [10]; patients whose AAA would not naturally rupture over the course of their lifetime may be put at unnecessary risk if their AAA were to be repaired. In light of these observations, the pathophysiology and biomechanics associated with AAA rupture are much more complicated than aneurysm diameter alone.

Aortic wall stress and aortic wall strength each play an important role in the biomechanics of AAA rupture. As a consequence, much work has been devoted to explore biomechanical relationships among stress, strength, and rupture risk. In particular, many studies have focused on quantifying peak wall stress (PWS), typically defined as the maximum stress occurring under systolic blood pressure (SBP) within the aortic wall. Finite element analysis (FEA) provides a means for calculating PWS on a patient-specific basis, utilizing clinical data already acquired as part of routine AAA evaluation: computed tomography (CT) scans and SBP. Moreover, studies have also focused on quantifying wall strength. The end goal of these efforts has been to utilize biomechanical-derived quantities as patient-specific clinical predictors of AAA rupture risk more effectively than current diameter-based criteria.

In this review, we highlight the main studies that have utilized biomechanical analysis and computations to assess the risk of patients with AAA. While the literature on AAA biomechanics is extensive, with numerous studies addressing several different aspects of AAA with varying degrees of complexity, many studies consider only one or a few patients. In this review, we focus on studies that included more than 10 patients, grouped by outcome (e.g., ruptured AAAs vs. non-ruptured AAAs). Specifically, we describe the application and development of computational modeling techniques, with particular emphasis on the outcomes and limitations of these studies. Lastly, we present our suggestions for the future course of studies investigating the relationship between biomechanics and AAA rupture. Before we get to the main point of this review, we preface our discussion with a brief background on AAA structure and introduce important information regarding common FEA modeling approaches employed in AAA research.

## 2. Biomechanics of Abdominal Aortic Aneurysm (AAA)

The abdominal aorta is the main arterial vessel supplying the lower extremities and abdominal viscera (see Figure 1). The aorta bifurcates into the right and left common iliac arteries just above the pelvis. While the abdominal aorta is typically about 20 mm in diameter, an aneurysm occurs when aortic diameter exceeds 30 mm. Virtually all FEA studies on AAAs focused on aneurysms that occur within the infrarenal segment of the abdominal aorta, beginning just below the renal arteries. Most infrarenal AAAs have an intraluminal thrombus (ILT) in between the wall and the lumen that affects AAA biomechanics, together with the biological structure of the AAA wall and blood flow in the lumen.

### 2.1. Aortic Wall

The normal aortic wall consists of three distinct layers: (1) tunica intima; (2) tunica media; (3) tunica externa (adventitia). The tunica intima is the innermost layer of the aortic wall, composed of a thin endothelial cell layer, a sub-endothelial layer, and connective tissue. The tunica media, the thickest portion of the aortic wall, resides in the middle. It is made of several alternating layers of elastic tissue (i.e., lamellae) and smooth muscle cells, which spiral around the vessel. Finally, the tunica externa is the outermost layer of the aortic wall, composed mainly of collagen and elastic fibers (not lamellae). Under physiological conditions, the tunica media and tunica externa are primarily responsible for bearing the hemodynamic load, the forces exerted by blood flow. The tunica media primarily provides distensibility, maintaining blood pressure, while the tunica externa provides a physical limit to expansion.

The anatomy of an AAA is markedly different than the normal abdominal aorta. AAA walls are frequently associated with degradation of the wall microstructure including the elastic lamellae, paucity of smooth muscle cells, and loss of the three distinct aortic layers [29,30], and thus, abnormal tissue properties [5]. Another common finding in pathological AAA wall tissue is the presence of localized calcified regions [31]. Calcifications, which typically occur within the tunica media, also change wall tissue material properties [32]. Not surprisingly, the presence of calcification has been shown to alter the distribution of AAA wall stress. However, reports conflict as to whether these alterations manifests as an increase [33,34] or decrease [35] in PWS and risk of rupture. The effects of aortic wall calcification are, however, relatively unexplored from a biomechanical perspective.

### 2.2. Intraluminal Thrombus (ILT)

The ILT is essentially a blood clot, predominantly composed of fibrin and blood cells, that is present in most AAAs [36]. ILT forms due to the characteristics of blood flow within an aortic expansion, including flow recirculation and sometimes tissue tearing. Approximately 75% of AAAs contain an ILT [37], although the distribution of ILT within an AAA can vary widely. Once an ILT has formed, it disrupts the normal mass transport between blood and the aortic wall. The ILT also harbors inflammatory cells and proteases that may degrade the tissue extracellular matrix, leading to decreased wall strength. ILT has, thus, been implicated in wall weakening (a reduction in wall strength) via localized hypoxia and inflammation [38,39]. Furthermore, disruption of normal hemodynamic wall shear stress stimuli on the wall endothelial cells, which are no longer in direct contact with the flow of blood, leads to loss of function and further wall weakening [40,41]. Mechanically, however, ILT provides an additional layer between the blood and the aortic wall that presumably diminishes wall stress [42,43,44,45,46]. The balance between wall stress and wall strength due to ILT formation might be an important—yet relatively unexplored—factor in AAA rupture.

### 2.3. Lumen

The aortic lumen is where blood flows. Lumen size depends on the aneurysm wall and ILT geometry. Hemodynamic forces within the lumen, particularly blood pressure, generate stress on the AAA wall, which, in tandem with aortic wall strength, may ultimately result in rupture. Other hemodynamic forces such as wall shear stress (WSS) have been shown to add little to rupture risk prediction [47]. Nevertheless, the geometry of the lumen plays an important role in the generation of wall stress, and thus, on AAA rupture.

## 3. Computational Modeling Techniques

Before delving into the details of AAA studies and their findings, let us take the time to briefly introduce some of the key concepts used in biomechanical analysis of aortic walls. We will then describe some of the methods and models employed for wall stress quantification, and their assumptions and implications. Finally, we will address models of aortic wall strength.

### 3.1. Basic Biomechanics Concepts and Aortic Wall Stress Quantification

Physically, mechanical stresses are forces (per unit surface) in a “body” or tissue. Because forces can act in all spatial directions, stress is mathematically expressed by a tensor (which in 3D can be represented as a 3 × 3 matrix, containing all nine spatially varying components of the stress tensor). The use of tensors allows us to properly extract information about forces applied to a specific plane or surface in the tissue. Following basic mechanical principles, when computing stresses, an equilibrium of forces (in all directions) is imposed. If we think of the aorta as a perfectly cylindrical blood vessel (an initial assumption), then the stresses exerted on the aorta by the internal blood pressure (P) have to be counteracted by the aorta tissue stresses (see Figure 2) so that forces are in equilibrium. In the circumferential direction, equilibrium of forces implies:(1)$$2\;P\;R=2\;\int_{R}^{R_{o}}\sigma_{\theta }\;dr$$
where *R* is the cylinder internal (lumen) radius, *R*_o_ is the cylinder external (aortic tissue) radius, and *σ_θ_* is the circumferential stress. Please note that if we assume that the aortic wall is thin and the circumferential stress across the wall thickness is constant, Equation (1) leads to the Laplace equation, which is frequently used to estimate wall stresses in cardiovascular applications,
(2)$$\sigma_{\theta }\;=\;\frac{P\;D}{2\;h}$$
where *D* is the vessel diameter (*D* = 2*R*) and *h* is the wall thickness (*h* = *R*_o_ − *R*). While a simplification, Equation (2) leads to the notion that the (circumferential) wall stress increases with internal blood pressure, vessel diameter, and reduced wall thickness.

The presence of an ILT, which grows on the inner aortic wall surface (the wall–lumen interface), changes the lumen geometry (see Figure 1 and Figure 2). In addition, if the ILT can carry some of the stress, then the total stress on the aortic wall decreases. Unlike cases with no ILT, blood pressure is applied to the thrombus and only indirectly to the aortic wall, and thus, the reduction in lumen radius/diameter also leads to a reduction in aortic wall stress.

The aorta, however, is not a perfect cylinder. Aortic wall stress, therefore, depends on the exact geometry of the AAA wall and thrombus. This includes the wall and thrombus thickness, and possible variations in thickness along the AAA. Stresses in all directions, for example, the circumferential and longitudinal directions, also need to be considered. To compute the maximum stress, the von Mises or principal stress, a local measure of maximum stress (which varies spatially over the AAA tissue), is frequently used. However, because aortic and AAA tissue is anisotropic, with stresses depending on the orientation of collagen fibers and the tissue microstructure, the direction of stress is also important to determine tissue tear. Aortic tissue can withstand higher stresses in the direction of collagen fibers than in a direction perpendicular to these fibers. Aortic wall tissue will tear if the wall stress is too high, and the tissue cannot withstand the forces that are trying to pull it apart. Consider the case of circumferential stresses depicted in Figure 2 and assume that collagen fibers are oriented in a circumferential direction. Circumferential stresses are pulling on the tissue to withstand the blood pressure, but if the blood pressure is too high, or the tissue too weak, the tissue will tear, leading to bleeding if the tear propagates through the wall thickness. Aortic rupture, thus, depends on both the wall stresses, which depend on the vessel geometry, microstructure, and internal blood pressure; and how strong the tissue is, which is quantified by wall strength, *σ**. The tissue will tear and the aortic wall will rupture if the wall stress exceeds the wall strength [5,48,49]. Thus, to estimate risks of rupture for a particular patient, we need to know or estimate both the AAA wall stresses and wall strength.

### 3.2. Quantifying Aortic Wall Stress

To account for the intricate geometry of the aortic aneurysm in computations of wall stress, FEA is typically employed. Initially, FEA studies were limited and assumed idealized AAA geometries [50,51,52,53]. Those initial studies, however, showed the effect of AAA geometry and tissue properties on wall stresses, guiding later studies. With advances in medical imaging and imaging analysis techniques over the past 20 years, generating three-dimensional (3D) image-based AAA geometries for FEA analysis from CT scans or MRI images of the AAA became feasible. Because CT scans are routine in AAA evaluation, studies involving FEA of image-based (patient-specific) AAA geometries do not subject patients to additional invasive procedures. Studies can be done retrospectively, allowing selection of patients based on outcomes or specific characteristics, as well as enabling larger cohorts to be analyzed.

Over time, the complexity of subject-specific biomechanical models of AAAs has increased. Pioneer studies in the field modeled the aortic wall using 2D shell elements (i.e., surface meshes) neglecting the effects of the ILT [54,55,56,57,58] and assuming constant wall stress through the wall thickness (like the Laplace model). These studies, however, were the first to establish the utility of employing biomechanics in the assessment of AAA rupture risk. Later studies recognized the importance of the ILT in AAA biomechanics, first merely as an additional layer that reduced wall stress, but then, with the recognition that ILT could also influence wall strength. Because of the more complex geometry, FEA models that included ILT employ 3D elements rather than 2D shell elements [4,59]. Most studies still assume that the aortic wall is of uniform thickness, an assumption that has been refuted by many anatomic cadaveric studies [60]. This approximation comes from limitations in the resolution of CT scans (usually around 0.5–1 mm per pixel), from which the aortic wall thickness (0.5–2 mm) cannot typically be resolved. Despite these challenges, recent studies have begun to include variations in AAA wall thickness, as well as the effect of calcifications in the aortic wall [61].

The stress experienced by the aortic wall is governed almost exclusively by hemodynamic conditions, in particular, blood pressure. Due to the pulsatility of blood pressure, wall stress within an AAA varies periodically with the cardiac cycle. However, because AAA rupture occurs when wall stress exceeds wall strength, rupture is most likely to occur at the time of highest load (i.e., systolic blood pressure, SBP) and at the point where wall stress is highest (the peak wall stress, PWS) and/or the wall is weaker (i.e., low wall strength). Thus, virtually all studies that have investigated AAA biomechanics employ static (steady-state) simulations, with most applying SBP at the luminal interface of the AAA as a (normal traction) boundary condition. For the remaining boundary conditions, the proximal and distal ends of the AAA wall, which are typically defined immediately distal to the renal bifurcation (proximal end) and either immediately proximal or immediately distal to the aortic bifurcation (distal end), are fixed in space to simulate tethering to the remainder of the cardiovascular system.

#### 3.2.1. Classical Model

To compute AAA wall stress, the material properties of the AAA wall and ILT need to be specified. Material properties relate the stress to the tissue deformation (e.g., tissue stretch) by means of a mathematical function. These properties can be linear or nonlinear. In the linear case, stresses are related to deformations, or strain, by a simple equation:(3)σ=E ε
where **σ** is the stress tensor, **ε** is the strain tensor (which contains information on how the tissue has deformed with respect to a stress free geometry or configuration), and **E** is a stiffness matrix (which specifies how stiff or distensible the tissue is). Except when deformed minimally, tissues usually have a nonlinear stress–strain (or stress–stretch) relationship and the equations that capture tissue material properties are generally more involved than Equation (3).

Several studies have been conducted to determine the material properties of the AAA wall and ILT using tissues from patients undergoing elective repair and/or from cadavers [48,62,63,64,65,66]. In particular, Raghavan et al., 2001 [62] developed a hyperelastic, isotropic material model of the AAA wall tissue, which is commonly employed in AAA biomechanical modeling efforts. The constitutive relations corresponding to this hyperelastic nonlinear AAA tissue material model are described by:(4)$$\sigma =-p\;I+2\;\left[\alpha +2\;\beta \;\left(I_{1}-3\right)\right]\;B$$
where **σ** is the Cauchy stress tensor, *p* is the hydrostatic pressure, **I** the identity tensor, **B** is the left Cauchy–Green tensor, and *I*_1_ is the first invariant of **B** (*I*_1_ = tr (**B**)). The model parameters (α = 17.4 ± 1.5 N/cm^2^; β = 188.1 ± 37.2 N/cm^2^) were determined from uniaxial tensile test data obtained from 69 human AAA specimens [62]. For ILT, the most common constitutive relation assumed is that of Wang et al., 2001 [64], which follows the same form as Equation (4), but has different parameters (α = 2.6 N/cm^2^; β = 2.6 N/cm^2^), and was derived from uniaxial data measurements from 50 ILT specimens. It is worth mentioning that measurements showed large variations in wall and ILT properties among patients and even in different locations for the same patient.

More recent studies showed that both the wall and ILT present anisotropies [36,48,65,67] (not accounted for in Equation (4)). Anisotropies are due to the tissue microstructure and in particular, the orientation of collagen fibers [68]. However, without excising the AAA tissues (which is not feasible in clinical practice before surgical repair), the distribution of AAA material properties in an actual patient are not known and cannot be inferred with current noninvasive techniques. Further, patient-to-patient variability is large [48,67]. Given this limitation, studies have typically modeled the wall and ILT as uniform and isotropic materials, using Equation (4), with the constants specified above. Nevertheless, recent studies have started to account for AAA tissue anisotropies, e.g., references [59,66,69].

Another factor to consider in AAA modeling is the so-called initial stress of the aneurysm. Material model equations are written with respect to an unstressed (stress free) configuration. Because patient CT scans are acquired in vivo, the images depict a strained (deformed) AAA geometry resulting from the applied internal blood pressure. In fact, during the cardiac cycle, the elasticity of the wall allows for expansion and contraction of the AAA. This can best be appreciated with ultrasound imaging. For CT scans, due to the time it takes for image acquisition, the obtained image is actually an average of the geometry over the cardiac cycle. Consequently, the 3D reconstructed AAA geometry based on these images is in equilibrium with blood pressure (mean arterial blood pressure) and is under stress conditions. The unstressed geometry/configuration is not known. This is frequently neglected, and many models assume the 3D CT scan geometry corresponds to an unstressed and unloaded configuration. However, neglecting this initial stress in FEA simulations can change the computed magnitude of wall stress by 21–42% [70,71].

Realizing these difficulties, some models first transform the CT scan geometry into an unstressed configuration [70] or compute the initial stresses [71]. Then, they use this configuration or initial stress as a starting point in the computation of wall stresses, imposing a blood pressure that deforms the AAA to the CT scanned geometry and then, quantifying wall stresses from the deformed (CT scan) geometry. Theoretically, this approach should be more precise at computing wall stresses. However, it faces several difficulties: 1. Because material properties are nonlinear (that is, the tissue behaves differently depending on how much it is stretched/deformed), it becomes mathematically and computationally difficult to find the unstressed initial configuration; 2. Residual stresses, which are stresses due to unknown longitudinal and circumferential stretch of the arterial tissues, are frequently ignored, even though they can be important contributors to arterial and AAA wall stress. Incorporating residual stress distributions makes the problem of finding the initial unstressed configuration even more intractable. Because of these difficulties, many of the studies, especially those that used relatively large number of patients (n > 10), neglected to account for these effects.

#### 3.2.2. Equilibrium Model

Rather than performing intractable computations, we can use basic equilibrium concepts in mechanics to estimate AAA wall stresses. One such example of how equilibrium conditions can be used to our advantage is the Laplace equation (Equation (2)). To compute circumferential wall stress using the Laplace equation, we have assumed that the walls were thin (so that stress is approximately constant throughout the wall). Note, however, that we did not have to make any assumptions about deformations or the initial configuration, nor even the material properties of the wall tissues. We simply used the mechanical principle of equilibrium of forces. Zelaya et al. [72] realized that this concept could also be used in the computation of AAA wall stresses, even when the walls are not thin and an ILT is present. They then used the AAA CT scan geometry to compute the wall stresses in equilibrium with the internal blood pressure.

If we were to use FEA to compute wall stresses, tissue material properties would need to be specified, and computations would result in tissue deformations. These deformations, however, can be tiny (negligible). To preserve the CT scan geometry, we can use a very stiff yet linear material model (Equation (3)) for the tissue, while imposing the patient blood pressure onto the intraluminal surface. This accomplishes the computation of the equilibrium wall stress under the applied blood pressure with negligible tissue deformations, thus, preserving the CT scan geometry [72]. To validate this approach, a series of in silico comparisons against ‘ground truth’ cases (in which the AAA tissues were assumed to be nonlinear and even having residual stresses, but starting from known unstressed configurations) were performed. Results showed that wall stresses computed using this new straightforward approach were remarkably accurate [72]. This simple concept was later adopted by other groups [73,74]. Thus, simple linear models can be used to accurately estimate AAA wall stresses under equilibrium conditions (under internal blood pressure loads) using CT scan geometries. Importantly, using a straightforward linear model, we can accurately estimate wall stresses without requiring any knowledge about the material properties of the AAA wall.

### 3.3. Models of Aortic Wall Strength

Because wall strength can vary, not only among patients, but also within an individual AAA [38,49], properly estimating wall strength is crucial in predicting patient-specific AAA rupture risk. Aortic wall strength is affected by many factors including sex, ILT distribution, family history, and genetics [49]. Typically, wall strength estimations are based on a single study that associated patient clinical factors with tensile strength testing of surgically collected AAA specimens [49]. The study included 60 uniaxial test samples obtained from 29 AAA patients and established the following relationship for aortic wall strength (*σ**),
(5)$$\sigma *=719-379\;\left(\sqrt{ILT}-0.81\right)-156\;\left(NORD-2.46\right)-213\;HIST+193\;SEX$$
where σ* is in units of kPa, *ILT* is the ILT thickness in units of cm, NORD is a dimensionless parameter for local normalized diameter, HIST is a dimensionless binary variable (1/2 for positive family history, −1/2 for negative family history), and SEX is a dimensionless binary variable (1/2 for males, −1/2 for females) [49]. This empirical relationship has been the basis for many patient-specific biomechanical studies that approximate aortic wall strength [4,42,74,75,76].

## 4. Studies and Limitations

We will focus here on reviewing the results and conclusions from relatively large (n > 10) AAA patient studies that used biomechanical analysis. Most of these studies, at least initially, used the nonlinear, albeit isotropic, material properties described by Equation (4), and later studies incorporated estimations of the aortic wall strength using Equation (5). Over time, biomechanical models became more involved, and included the intraluminal thrombus (initially neglected), residual and initial wall stresses, as well as the effects of calcifications. More recently, computational models have also incorporated tissue anisotropies. Nevertheless, biomechanical models have not been broadly adopted in clinical care of patients with AAAs. We will explore the limitations that preclude current clinical implementation of biomechanical models of AAA.

### 4.1. Initial Studies

The first two relatively large patient studies to investigate the relationship between biomechanics and AAA rupture were conducted by Fillinger et al. [54,55], and soon, other studies followed [56,57,58]. In these studies, the outer contour of patient-specific AAA geometries was extracted from patient CT scans and modeled using shell elements in FEA, assuming constant wall thickness, and the material properties described by Equation (4). Patient-specific systolic blood pressure (SBP) was applied as the internal AAA blood pressure load. Results from these initial studies showed that peak wall stress (PWS), the maximum principal wall stress in the model, correlated with AAA rupture better than AAA maximal transverse diameter (see Table 1). These studies were the first to establish PWS as a predictor of AAA rupture risk.

Each of these initial studies further contributed diverse aspects to the biomechanical analysis. In the first study by Fillinger et al. [54], PWS was calculated for a total of 48 patients (30 electively repaired AAAs, 8 symptomatic AAAs, and 10 ruptured AAAs). Results showed that the ruptured/symptomatic AAA groups exhibited significantly higher PWS compared to the electively repaired AAA group. These results were later confirmed by Heng et al. [58], using a total of 70 patients (40 electively repaired AAAs and 30 acutely repaired AAAs): PWS was significantly higher for AAAs undergoing acute repairs compared to those undergoing elective repairs. The study by Fillinger et al. also highlighted the importance of using patient-specific SBP in calculating PWS: using a normalized SBP (e.g., 120 mmHg) reduced the significance of findings [54]. A later study by Truijers et al. [57], which included a total of 30 patients (10 asymptomatic AAAs, 10 symptomatic AAAs, and 10 ruptured AAAs) and focused on patients with small AAAs (with maximal diameters < 55 mm), confirmed findings that PWS is a better predictor of AAA rupture when SBP is used as the load. This observation has clinical implications, as hypertension is regarded as a risk factor associated with AAA rupture [77], and patients with ruptured/symptomatic AAAs tend to have higher SBP than patients with non-ruptured AAAs [55,56,57].

A second study by Fillinger et al. [55] further investigated the relationship between PWS and AAA rupture. In this retrospective study, PWS was calculated using geometries extracted from initial presenting CT scans; and patient outcomes over the following year were assessed from the medical record. The study included 103 patients (42 observed AAAs without intervention, 39 electively repaired AAAs, 8 symptomatic AAAs, and 14 ruptured AAAs). Results from this study showed that PWS was a more sensitive criterion than maximal diameter for distinguishing between patients who became symptomatic/ruptured and those who were asymptomatic and underwent elective repair, significantly differentiating the two size-matched groups. Moreover, for the ruptured AAA group, the location of PWS correlated with the location of AAA rupture. Together, study results began to establish PWS as a superior predictor of AAA rupture compared to maximal diameter alone.

A later study by Venkatasubramaniam et al. [56], which included a total of 27 patients (15 non-ruptured AAAs and 12 ruptured AAAs), corroborated that the location of PWS correlated with the location of AAA rupture. Furthermore, by performing a subset of simulations that parametrically adjusted AAA wall thickness, this study also showed that under the same conditions, PWS was inversely related to aortic wall thickness. While this is not surprising (see e.g., Equation (2)), the study brought attention to the biomechanical importance of wall thickness. Unfortunately, estimation of AAA wall thickness continues to be limited by the accuracy of CT scans, which frequently cannot resolve the AAA tissue wall thickness and its variations in space.

Despite exciting results, these initial studies share an important limitation: the FEA model failed to account for the biomechanical effects of the ILT, and wall strength was not considered. Although there was a degree of uncertainty in the literature at the time regarding the importance of ILT in AAA biomechanics [50,62,78,79,80,81], subsequent studies showed that the ILT can significantly affect the distribution of wall stress and lead to reductions in PWS [42,43,44,45,46]. By neglecting the effects of ILT on wall stress, PWS in these studies was highly influenced by AAA geometry, including diameter, and SBP. Furthermore, the models did not consider possible changes in wall strength, effectively assuming common wall strength among patients that would justify PWS measurements as predictive of rupture. The results from these five studies should, therefore, be interpreted with a clear acknowledgement and understanding of their limitations.

### 4.2. Studies Accounting for the Intraluminal Thrombus

Following several studies that emphasized the possible effects of ILT on AAA tissues, patient studies started to incorporate the effects of ILT on wall stress. To this end, and because the distribution of ILT varies along the circumference and longitudinal length of the AAA (e.g., see Figure 1), 3D FEA models were used (i.e., 3D elements rather than 2D shell elements). These elements, further, can assess changes in wall stress through the AAA wall and ILT.

Initial patient studies (n > 10 patients) incorporating ILT in FEA models were performed by Vande Geest et al. [4,59]. These studies used Equation (4) to model the properties of the ILT [64]. The first of those studies, published in 2006 [4], included a total of 13 patients (5 non-ruptured AAAs and 8 ruptured AAAs). Unlike previous studies, they utilized a hyperelastic anisotropic material model with preferred stiffening in the circumferential direction to describe the AAA wall [63]. The second study, published two years later in 2008 [59], included a total of 35 patients (21 electivity repaired AAAs, 5 non-ruptured AAAs, and 9 ruptured AAAs). This later study further compared the efficacy of the hyperelastic anisotropic AAA wall model [48], to that of the widely used isotropic model used previously and highlighted in Equation (4) [62]. Results demonstrated that within the electively repaired AAA group, the anisotropic model produced significantly higher PWS compared to the isotropic model, both with and without ILT, and further showed that simulations including ILT resulted in significantly lower PWS than those that neglected the ILT [59].

Interestingly, despite improved model assumptions, these two studies, which incorporated the ILT and utilized more realistic wall material models, also showed non-significant increases in PWS in ruptured AAAs [4,59]. The results contradict the conclusions of the initial five relatively large (n > 10) patient studies [54,55,56,57,58] highlighted above, all of which showed significant increases in PWS, albeit excluding ILT from their models. This contradiction could be explained by a smaller number of patients sampled by Vande Geest et al. [4,59], or the actual incorporation of ILT in their models that resulted in lower PWS. The studies, nevertheless, demonstrated the biomechanical importance of ILT in reducing wall stress and therefore, PWS.

Because in the studies by Vande Geest et al. PWS was no longer a good predictor of AAA rupture, wall strength, which was neglected before in the stratification of patients, started to be included in models [4]. To compute wall strength, the strength statistical model developed previously [49], Equation (5), was used. Results showed a significant decrease in wall strength in the ruptured AAA group. The decreased wall strength of ruptured AAA specimens compared to non-ruptured AAA specimens was further demonstrated by the same group in a clinical study [82].

Regardless, when PWS and wall strength were combined as a rupture potential index (RPI), defined as the ratio between PWS and wall strength, differences between the ruptured and non-ruptured AAA groups did not reach statistical significance. However, the *p*-value associated with RPI was lower than for comparisons of PWS and diameter alone, suggesting that RPI may be a better predictor of risk. The authors, thus, suggested that RPI will likely reach statistical significance between ruptured/non-ruptured groups, effectively stratifying patients, if the number of patients in their study were to be increased. Wall strength, a component missing from previous studies, in combination with wall stress through RPI, thus, emerged as an important factor in rupture risk prediction [4]. Subsequent studies followed this idea, moving away from the assumption that AAA rupture is dictated predominantly by PWS, and instead, assuming both wall stress and wall strength are involved [42,75,76].

### 4.3. Refined Models of AAA and RPI

Subsequent patient studies, which generally incorporated the ILT, showed a divergence in results. These studies continued to expand on many of the developed models and techniques, including the refinement of PWS-derived indexes such as the RPI, incorporation of patient-specific AAA wall thickness, and inclusion of AAA wall calcification. However, while some studies showed a significant increase in PWS or RPI in the ruptured and symptomatic AAA groups compared to non-ruptured groups, some studies did not find such increase.

Studies by Gasser et al. [42], considering diameter matching (n = 35; 18 ruptured) and diameter-pressure matching groups (n = 42; 18 ruptured), compared different modeling approaches. Instead of using patient-specific SBP, however, they used mean arterial pressure (MAP) in their models, and simulated models in which AAA wall thickness was dependent on MAP and/or ILT thickness. Most interestingly, in their study, the models including ILT showed significant increases in PWS and RPI (which they called instead peak wall rupture risk, PWRR), whereas the models that excluded ILT did not show a significance increase in PWS, contradicting previous studies. Overall, however, this study highlights the critical role that modeling assumptions can play in calculating PWS and RPI, and thus, on using these models on actually predicting patient risks.

The same year (2010), a study conducted by Maier et al. [75] included a total of 53 patients (with 14 ruptured AAAs). Unlike previous studies, this study incorporated the effects of the initial stress state on the AAA. Results showed that PWS was significantly greater for the ruptured/symptomatic AAA group compared to the non-ruptured AAA group; RPI, however, displayed a more pronounced increase on a size-matched comparison. The study, therefore, highlighted the potential of RPI over PWS in predicting rupture risk for AAAs. These results and the advantage of RPI over PWS in stratifying patient risks were confirmed in a later study performed by Erhart et al. [76], which included a total of 60 patients (15 ruptured AAAs). In a later study, Shang et al. [61] (n = 26, but only symptomatic and asymptomatic groups were considered) captured patient-specific aortic wall thicknesses directly from CT scans using sophisticated image processing techniques [83,84]. Moreover, their FEA model also included areas of AAA wall calcification, employing a calcific material model developed in a previous study [35]. Results from this study showed that PWS was significantly higher for the symptomatic AAA group compared to the asymptomatic AAA group, but only when using patient-specific wall thickness.

As studies continued to become more involved, including different wall material properties, the effects of ILT, wall thickness variations, different blood pressure loads, and wall calcification, results from FEA have also became more contradictory and not quite reproducible among groups. One common problem among studies is that many of them used images from ruptured AAAs, which may not properly reflect the pre-rupture status needed for meaningful quantifications. While some studies continue to embrace the notion that PWS could be used as a superior predictor of AAA rupture risk, RPI (and PWRR) have emerged as better risk predictors. The fundamental idea of considering both wall stress and wall strength as criteria for AAA rupture makes sense from a biomechanics perspective. However, while wall stress calculation seem to have become more accurate, patient-specific wall strength estimations remain elusive, given that tissue and ILT material properties, and wall thicknesses, vary widely from patient to patient and spatially within the same patient [5]. More refined computational models and experimental systems to investigate the effects of blood flow and flow pulsatility (not just blood pressure), and even vascular growth and remodeling have also been implemented [85]. Geometrical changes have been investigated [86] as well as geometry-derived indexes, including tortuosity and surface curvature [87,88,89,90]. However, biomechanical indexes of rupture continue to be elusive and rupture mechanisms are not yet fully understood. Model assumptions could therefore play a large role on conclusions made, with wrong conclusions potentially generated on a patient-specific basis following incorrect assumptions based on grouped data.

Indeed, studies are starting to question the importance of biomechanical analysis in AAA. A recent study [91] (which included 175 asymptomatic, 11 symptomatic, and 45 ruptured aneurysms), found that after diameter matching, there were no significant differences among groups in PWS or RPI. More recently, a retrospective study was conducted on 13 patients with ruptured AAAs, who were imaged before and after rupture [92]. The AAA software employed computed the rupture risk equivalent diameter (RRED), an index based on biomechanical quantifications (PWS and RPI, see [93]). The study found that RRED was actually smaller than the actual AAA diameter (suggesting a lower risk of rupture) in more than half of the pre-ruptured AAA cases. The study concluded that biomechanical indices are not yet ready to stratify patients for AAA rupture risk and thus, are not ready for clinical practice. More research is needed to address important gaps in our understanding of AAA biomechanics and how they affect rupture before effective clinical translation is possible.

AAA is characterized by degradation of the aortic wall tissue (or remodeling), including loss and fragmentation of elastin, fibrillin fragmentation, increased collagen content accompanied with decreased alignment of collagen fibers, and loss of smooth muscle cells [1,5,66,94]. This degradation affects the mechanical properties of the AAA wall, including its strength, and it varies significantly from patient to patient as well as within an AAA [5]. Because there is no current clinical test to non-invasively assess microscopic tissue composition, AAA wall properties continue to be elusive [95]. Attempts have, therefore, been made to correlate AAA wall properties, including wall strength, to non-invasive measurements such as metabolic activity (assessed by positron emission tomography scans, PET/CT) or blood biomarkers of tissue degradation activity, e.g., [94,96]. More data, however, are needed to further validate results and achieve a more reliable assessment of AAA wall properties including strength.

Advanced computational biomechanical models that incorporate the microstructure of the AAA wall are emerging. The large variability of tissue mechanical properties in AAA patients is due to changes in the AAA tissue microstructure that occur due to degradation [68]. By looking at the relationship among microstructure and mechanical properties, different stages of AAA tissue remodeling progression could be identified, including a substantial isotropic stiffening of AAA walls [66]. Biomechanical models can nowadays incorporate information on tissue microstructure, most notable on collagen fiber orientation, and thus, reproduce the behavior of the AAA wall in silico [66,68]. AAA wall tissue remodeling certainly influences the strength of the AAA, contributing to rupture risk. The relationship between tissue strength and microstructure, however, needs to be fully investigated. Furthermore, growth and remodeling (G&R) biomechanical models are also emerging [97,98]. In the context of AAA, G&R models aim at predicting both the geometrical grow of the aneurysm as well as the accompanying tissue remodeling, with the dual goal of better understanding the progression of AAA and assessing patient risks. Advances in imaging modalities that can non-invasively identify tissue microstructure, adipocyte concentration, inflammation, and other details of the AAA wall, to assess degradation, together with advanced biomechanical models [99], are promising to help in future assessments of patient-specific rupture risks.

### 4.4. Beyond Biomechanical Models

Since biomechanical rupture indices are elusive, studies are starting to unravel situations that increase the risk of AAA rupture. While some patient underlying conditions and history (sex, smoking status, hypertension, genetic predisposition) are known to disproportionately affect patients, other conditions affecting rupture continue to emerge [100]. For example, Crawford et al. [100] demonstrated that reductions in aortic outflow from iliac occlusive disease are associated with increased peak wall stress and rupture of AAAs at smaller sizes. These studies are supported by historical data that demonstrated higher AAA rupture rate in patients with surgical ligation of the iliac arteries, which was done in combination axillo-bifemoral bypass as an experimental treatment for AAA, as well as in patients with reduced aortic outflow due to previous lower extremity amputations [101,102]. Moreover, Haller et al. [103] found that increased ILT burden is associate with an increased risk of rupture in small AAAs (<6 cm). Presumably, while the ILT decreases PWS, it also degrades the wall, reducing its wall strength. This finding opens the possibility of using ILT burden as a surrogate marker of rupture risk. Other surrogate markers are likely to emerge to complement biomechanical data, but are currently in preliminary stages.

One promising approach is the use of fibrillin degradation fragments in peripheral blood [94]. Fibrillin is a structural component of the aortic extracellular matrix that is closely associated with elastic lamellae. A defect in the Fibrillin-1 gene is also the underlying mutation in Marfan’s syndrome, which is associated with increased prevalence of aortic aneurysms. The underlying assumption is that degradation of fibrillin-1 will result in various fibrillin-1 fragments that can be detected in peripheral blood samples, with more elevated fibrillin-1 fragment levels correlating with advanced degradation of the aortic wall and decreased wall strength. Initial studies are promising in showing that fibrillin fragments are detectable in peripheral blood samples of patients with aortic aneurysms and dissections. However, this work is in the preliminary stages and studies to determine appropriate circulating values for patients without aortic aneurysms, as well as longitudinal variation in fibrillin fragment levels in patients with aortic aneurysms, are ongoing.

In addition to surrogate markers of aortic wall strength in peripheral blood, advanced imaging modalities, such as PET/CT scans, may improve predictions of rupture risk by providing information regarding the biological activity of cells within the aortic wall and degree of inflammation within the AAA. Some studies have suggested an association between increased activity on PET/CT scans, AAA growth, and increased PWS [104,105]. However, incorporation of PET/CT scan into rupture calculations is still of unknown utility, with a recent systematic review calling into question the potential improvement in rupture risk prediction offered by PET/CT scans [106].

## 5. Looking Forward

Current biomechanical models of AAA face uncertainties in quantifications of risk indexes that preclude their use in clinical practice. This is because patient tissue properties, in particular wall strength, cannot be properly inferred with current technologies and can only be vaguely estimated. Tissue strength varies significantly within the AAA and from patient to patient, making the RPI, even if PWS is accurately quantified, a crude estimate of risk. Further, changes in AAA wall thickness, which influence PWS, are difficult to extract from many clinical CT scans, which minimize exposure at the expense of accuracy [107]. Not surprisingly, studies are concluding that current biomechanical models are not yet ready to stratify patients. Thus, even though maximal transverse diameter is an inaccurate predictor of rupture risk, it continues to be used in clinical practice due to simplicity and because it may not be much less accurate than other methods.

We need better solutions for risk stratification of AAA patients. The key to stratification is likely a combination of biomechanical (wall stress, tissue microstructure) and biochemical surrogate markers (e.g., blood markers, metabolic activity, gene expression) [96,108], together with other risk variables (e.g., sex, smoking habits, underlying conditions). Artificial intelligence (AI) is revolutionizing healthcare. We envision that the use of AI will also revolutionize AAA management and risk prediction; see e.g., [109] for a review of current studies. By accessing large cohorts of patients (combining databases from hospitals and centers), it is possible to use artificial intelligence methods, such as machine learning and deep learning algorithms, to start unravelling associations among rupture risk, risks associated with repair (mortality but also complications such as endoleaks), and other patient variables (including biomechanical indices). Because AAA rupture is a biomechanical event, the work performed to unravel biomechanical factors of AAA rupture remains essential. However, biomechanics alone may not be the next predictor of AAA risk, as once promised, given all clinical uncertainties and limitations. Instead, we envision biomechanics as a key component of more sophisticated patient-specific models that incorporate diverse information.

## Figures and Tables

**Figure 1 bioengineering-07-00079-f001:**
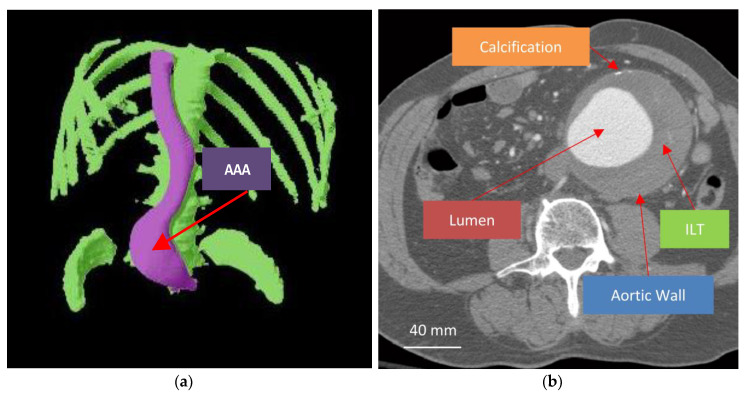
Abdominal aortic aneurysm (AAA). (**a**) Three-dimensional rendering of the aorta, with an AAA, and the vertebra, rib cage, and hip bones for reference. (**b**) Annotated transverse CT scan section illustrating the AAA lumen, intraluminal thrombus (ILT), and aortic wall with calcification. Note: from this CT image, aortic wall thickness cannot be determined.

**Figure 2 bioengineering-07-00079-f002:**
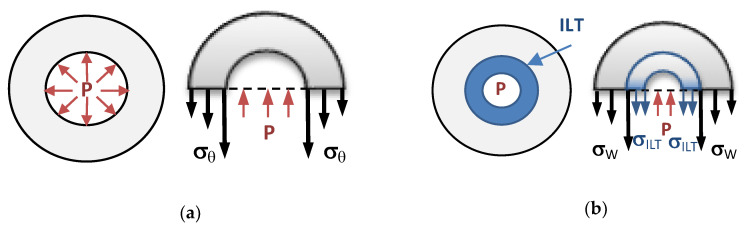
Sketch of stresses acting on a perfectly cylindrical blood vessel. (**a**) Cylindrical vessel wall (gray) subjected to an internal blood pressure (*P*). If we cut the cylinder in half as shown in the right sketch, then equilibrium of forces requires that the pressure is equilibrated to the circumferential wall stress (*σ_θ_*), as seen in Equation (1). (**b**) Cylindrical vessel wall (gray) with intraluminal thrombus (ILT, depicted in blue). The pressure force is equilibrated by both circumferential wall stresses in the wall (*σ_W_*) and the ILT (σ_ILT_), thus lowering the wall stress.

**Table 1 bioengineering-07-00079-t001:** Initial pioneering biomechanical studies on AAA. These studies have all used patient-specific AAA geometries extracted from CT scans, and modeled the AAA using shell elements in FEA. They further assumed constant wall thickness, the material properties described by Equation (4), and did not include an intraluminal thrombus (ILT).

Study	AAA Patients (n)	Findings
Fillinger 2002	48 (30 electively repaired; 8 symptomatic; 10 ruptured AAAs)	PWS correlates with AAA rupture; patient-specific SBP needed
Fillinger 2003	103 (42 observed AAAs without intervention; 39 electively repaired; 8 symptomatic; 14 ruptured AAAs)	Location of PWS correlated with location of rupture
Venkatasubramaniam 2004	27 (12 ruptured)	PWS inversely related to wall thickness
Truijers 2007	30 (10 asymptomatic; 10 symptomatic; 10 ruptured AAAs)	PWS correlated with AAA rupture; patient-specific SBP needed
Heng 2008	70 (40 electively repaired; 30 acutely repaired AAAs)	PWS significantly higher for patients undergoing acute repair

PWS—Peak wall stress; SBP—systolic blood pressure.

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
