# Peer review of "Predictors of Abdominal Aortic Aneurysm Risks"

_bioengineering, 2020, doi:10.3390/bioengineering7030079_

Round 1

Reviewer 1 Report

     The authors summarize investigations of Biomechanical mechanisms associated with rupture of abdominal aortic aneurysm (AAA). Clinical decision-making currently prioritizes maximal aortic diameter as the primary factor in considering elective surgical repair in asymptomatic patients.

     Predictions based upon measures of peak wall stress stimulated initial interest in additional factors predictive of rupture risk. Unfortunately, enthusiasm waned as other measurements were added. The use of patient specific wall strength was promising, but also foundered as more measurements were added to the equations. The predictive equation for wall strength was based on a single biomechanical study in 2006 which has not been replicated. The authors have added measurement of intraluminal thrombus based on their JVS publication in 2018. Calcification is mentioned but engendered minimal discussion.

     Wall thickness would probably be useful but CT measured wall thickness is not accurately rendered by CT. They conclude that the predictive value of biomechanical determinants is not ready for clinical use, and seem pessimistic about further biomechanical determinants.

     These comments refer to pagination derived from the electronic review version and sequentially numbered lines.

  1. INTRODUCTION, p1, line 31-32: You state: “Aneurysm rupture, which occurs when aortic wall stress exceeds aortic wall yield strength, represents the main concern associated with AAA, as AAA rupture carries an overall mortality rate of approximately 80-85% [2, 3].” The first half of this sentence suggests that rupture is due to a balance between aortic wall stress exceeding aortic wall yield strength. Do either of these 2 references support this hypothesis? Could the authors define this with a more structured reference that specifically relates these 2 parameters to AAA rupture. If this information is not available this needs to be adjusted in the text. Is this a known fact or a biomechanical hypothesis?
  2. INTRODUCTION, p1, line 31-32: You state: “Aneurysm rupture, which occurs when aortic wall stress exceeds aortic wall yields strength, represents the main concern associated with AAA, as AAA rupture carries an overall mortality rate of approximately 80-85% [2, 3].” The two references provided (#2 and #3) both date from the 1980s. Mortality from ruptured AAA is still high but it probably does not remain at the level of 80-85%. Could the authors refresh this information with a more recent mortality estimate?
  3. INTRODUCTION, p1, line 36-38: This discussion of surgical repair includes estimates of in hospital mortality. You provide two references for this. One is an RCT published in 2010 and the second an article summarizing the US experience with aneurysm repair 20 years previously. A discussion of the risks of surgical repair is particularly important in a non-vascular Journal. However, the 21st century represents a new era in the treatment of AAA; there have been significant improvements in availability of tests (i.e. CT, US, MR, etc), technical improvements in tests, maturation of endovascular therapies, and improved clinical care to cite a few. Could the authors provide more recent citations supporting this consideration?
  4. Why do you complicate this presentation by introducing the comparison between open or endovascular repair? The important consideration is to balance the risk of elective repair against the risk of rupture. There are four RCTs available which report mid-term mortality--figures which are not restricted to 30 day or in hospital data. One of these (ACE) closed prematurely. DREAM and OVER have both published mid- and long-term results which are relevant to this discussion. Mortality in each of these trials equilibrated after 2 years and the 2 procedures had no difference in survival. Would these figures be more useful for the readership of this journal?
  5. INTRODUCTION, p2, line 41-48: A discussion of rupture in “small” AAA is interesting and clearly serves your proposition that aortic diameter should not be the only measure of rupture risk. However, the four articles cited were all published before 2000. This is an uncommon event and may be decreasing in incidence. The resolution and availability of both CT and ultrasound has improved considerably since 2000 making early diagnosis more likely. A recent RCT designed to demonstrate the benefit of early repair in smaller aneurysms failed to demonstrate a risk benefit to early repair. Should you include this in the discussion to balance the older articles? Are more recent studies available?
  6. Biomechanics of AAA P2, line 74: The first sentence in the introduction defines a AAA as an aorta with >3cm diameter supported by reference #1. You describe the same fact referenced by 2 separate references (#15 and 16) in this section. Why were more than one reference needed to support the definition of aneurysm? Or, why weren’t all three noted in the introduction?
  7. Biomechanics of AAA P3, line 81: You describe a “the vertebrates, rip cage.” Did you mean “vertebra, rib cage”?
  8. Computational modeling techniques, P6-7, lines 244-252: You describe the evaluation of the AAA during CT examination. The elasticity of the arterial wall permits expansion and contraction during the cardiac cycle. This may be one explanation for the difficulty in accurately measuring the thickness of the aortic wall. Can these computational analyses be reliably be calculated based on the largest or smallest diameter but not the random diameter at the time of slice acquisition? How is this adjusted in the calculation of these modeling calculations?
  9. Computational modeling techniques, models of aortic wall strength page 1-292-297. You report that all of these measures derived from a single 2006 study. Do we assume that no further corroboration of this work has been attempted? Why has so much subsequent work been based on a single unvalidated scientific study?
  10. REFERENCES: Key references supporting the assumptions inherent in AAA risk prediction are based on clinical evaluations and results from the previous century. Are there more recent studies that would provide more current information?
  11. Looking Forward, page 13 line 530: The authors state: “However biomechanics alone will likely not be the next predictor of AAA risk, as once promised, given all the clinical uncertainties and limitations. Instead we envision biomechanics as a key component of more sophisticated patient-specific models that incorporate diverse information.” This is a sober assessment of the extensive research evaluating biomechanical factors associated with the natural history of abdominal aortic aneurysm. If the major component of patient specific assessment is measurement of aortic wall strength, do the authors believe that additional studies of this measurement are essential to patient specific risk assessment? Should additional studies be done to evaluate the measurement of wall strength?
  12. Additional comment: The authors should clarify whether the measurements used in the CTAs were elective and preceded the event of aneurysm rupture. Were some of these studies done on the emergent CTs diagnosing rupture? CT or a ruptured aortic aneurysm may not reflect pre-rupture status and would further obfuscate reliability of these measurements.
  13. Additional comment: Aortic diameter is the most widely used clinical parameter associated with rupture, although elongation is frequently associated but rarely assessed in these investigations. Elongation and kinking at fixation points has become an important consideration affecting eligibility for secure Endovascular repair (EVAR). Has the Biomechanical research community addressed this parameter in relation to rupture?

Author Response

Please see the attached file. In the manuscript changes are marked using track changes.

Reviewer 2 Report

This is a well-written paper on predictors of abdominal aortic aneurysm (AAA) risks. Although the paper is very easy to read several aspects are missing or not up-to-date – several references are quite old (see, e.g., Table 1), and a lot is missing on more recent contributions, see, e.g., the recent review article on wall failure with a focus aneurysms (Acta Biomaterialia, 99:1-17, 2019). As we know now, the microstructure plays a significant role in the risk of rupture – the authors are invited to read the two articles describing this area: Acta Biomaterialia, 88:149-161, 2019 and Journal of the Royal Society Interface, 13:20160620, 2016. Aspects of G&R are completely missing, see, e.g., Computer Methods in Applied Mechanics and Engineering, 352:586-605, 2019 and Journal of Biomechanical Engineering, 137:031008, 2015, and aspects on mechanobiology are very relevant and need to be mentioned (see the review article published in Journal of Biomechanics, 45:805-814, 2012).

Some minor points:

Page 4: a tensor is not a 3x3 matrix in 3D.

Page 5: there is a typo in the “von Mises stress”. Please note that this type of stress measure is not meaningful for the analysis of AAAs.

Section 3.2.1:

- Hooke’s law is not meaningful, and should therefore be omitted.

- The isotropic law (4) is often used but several data do not show an isotropic response of AAA tissue. Hence, it is key to also add more recent papers, which use anisotropic laws, in particular in the last years. For the influence of material anisotropy on the mechanical AAA response see, e.g., ASME Journal of Biomechanical Engineering, 130:021023, 2008.  For aspects on constitutive modeling see the study in Journal of the Mechanical Behavior of Biomedical Materials, 41:92-107, 2015. The invariant I_1 is a scale and not a tensor, hence do not write the label I_1 in bold. B is not a stretch tensor, it is the left Cauchy-Green tensor.

Author Response

Please see attached file. In the manuscript changes are marked using track changes.

Round 2

Reviewer 2 Report

The authors have clearly made a sincere and thoughtful effort to respond to my comments. I congratulate them on their review and appreciate the diligence with which they have worked to improve the model. I recommend that the paper now be accepted for publication.